# A mechanism to prevent production of reactive oxygen species by *Escherichia coli* respiratory complex I

Marius Schulte[1], Klaudia Frick[1], Emmanuel Gnandt[1], Sascha Jurkovic[1], Sabrina Burschel[1], Ramona Labatzke[1], Karoline Aierstock[1,2], Dennis Fiegen [2], Daniel Wohlwend[1], Stefan Gerhardt [1], Oliver Einsle[1,3] & Thorsten Friedrich [1]

Respiratory complex I plays a central role in cellular energy metabolism coupling NADH oxidation to proton translocation. In humans its dysfunction is associated with degenerative diseases. Here we report the structure of the electron input part of *Aquifex aeolicus* complex I at up to 1.8 Å resolution with bound substrates in the reduced and oxidized states. The redox states differ by the flip of a peptide bond close to the NADH binding site. The orientation of this peptide bond is determined by the reduction state of the nearby [Fe-S] cluster N1a. Fixation of the peptide bond by site-directed mutagenesis led to an inactivation of electron transfer and a decreased reactive oxygen species (ROS) production. We suggest the redox-gated peptide flip to represent a previously unrecognized molecular switch synchronizing NADH oxidation in response to the redox state of the complex as part of an intramolecular feed-back mechanism to prevent ROS production.

[1] Institut für Biochemie, Albert-Ludwigs-Universität Freiburg, Albertstr. 21, 79104 Freiburg, Germany. [2] Boehringer Ingelheim Pharma GmbH & Co. KG, Lead Identification and Optimization Sup, 88397 Biberach, Germany. [3] BIOSS Centre for Biological Signalling Studies, Schänzlestrasse 1, 79104 Freiburg, Germany. Correspondence and requests for materials should be addressed to T.F. (email: Friedrich@bio.chemie.uni-freiburg.de)

The energy-converting NADH:ubiquinone oxidoreductase, respiratory complex I, is the first enzyme complex of most respiratory chains[1–7]. It is a major source of reactive oxygen species (ROS)[8], implicating a role in ageing[9], and its dysfunction was suggested to be related to human neurodegenerative diseases[10,11]. Despite the recently published structures of the complex[12–15], its mechanism, particularly concerning the connection of NADH oxidation and ROS production, is not fully understood. Complex I couples the transfer of two electrons from NADH to ubiquinone (Q) with the translocation of four protons across the membrane[16]. In doing so, it contributes to the generation of a proton-motive force required for energy-consuming processes. The mitochondrial complex of eukaryotes is composed of up to 44 different subunits, seven of which are encoded in the mitochondrial genome[1,4,7]. Most prokaryotic homologs consist of 14 different subunits, providing the structural core for an energy-converting NADH:ubiquinone oxidoreductase[3,6,12]. The L-shaped complex is structured into a peripheral arm that protrudes into the aqueous milieu and catalyzes electron transfer with one FMN and 8–10 iron–sulfur ([Fe–S]) clusters, and a membrane arm embedded in the lipid bilayer that mediates proton translocation[12–15].

The structure of the peripheral arm from the complex of *Thermus thermophilus* was determined to 3.1 Å resolution, revealing the arrangement of cofactors and suggesting an electron transfer pathway[17] (Fig. 1b). The electron transfer starts with NADH reducing the FMN via hydride transfer[18]. Subsequently, the electrons are passed on in several one-electron transfer steps over a distance of 95 Å along a chain of seven [Fe–S] clusters to ubiquinone[19]. For a discussion about the nomenclature of the [Fe–S] clusters see refs. [2,5,20]. An eighth cluster, called N1a, is located proximally to FMN, but on the side opposing the long chain of clusters. An electron residing on N1a may thus enter the pathway to Q only via the FMN, after re-oxidation of at least the first cluster of the chain, N3[21]. Here, we show that the redox state of N1a regulates NADH oxidation in complex I by inducing a conformational change close to the NADH-binding site. When N1a is reduced, the reaction product, NAD$^+$, cannot be ejected from its binding site and blocks the site preventing further ROS production.

## Results

**Structure of oxidized and reduced NuoEF.** We have determined structures of the electron-input module of complex I from the hyperthermophilic bacterium *Aquifex aeolicus*, encompassing subunits NuoE and NuoF that contain the FMN cofactor, the [4Fe–4S] cluster N3, and the [2Fe–2S] cluster N1a, respectively[22], both in the air-oxidized and dithionite-reduced states at resolutions up to 1.8 Å (Fig. 1; Supplementary Table 1). Crystals of *A. aeolicus* NuoEF contain two NuoEF heterodimers per asymmetric unit. Both monomers align with a root-mean-squared deviation (RMSD) of 0.17 Å for all atoms and will consequently not be discussed separately. NuoE and NuoF are very similar in structure to their homologues in *T. thermophilus*, Nqo1, and Nqo2, with a RMSD of 0.93 Å for 334 C$_\alpha$ atoms, in accordance with their high-sequence identity (Supplementary Figs 1, 2). However, our model of NuoEF contains 985 water molecules that are not detectable in the structure of the *T. thermophilus* complex I due to its lower resolution. The water molecules play an important role in NADH oxidation (see below). Subunit NuoF binds the FMN cofactor as well as cluster N3 and provides the binding site for NADH. The model for NuoF contains 418 amino acids, whereby eight residues at the C-terminus and the subsequent hexahistidine affinity tag at this position were not defined in the electron-density maps. The N-terminal domain of NuoF features a Rossman fold, albeit with

an uncommon mode of cofactor binding. Rather than at the variable loop regions C-terminal of the parallel β-sheet, the FMN moiety resides next to the sheet, located at the N-terminal end of the α-helix that connects strands 3 and 4. The resulting conformation of the cofactor thus differs substantially from what is commonly observed in flavodoxins. FMN resides at the protein surface and forms the major part of an extended, solvent-accessible cavity where the substrate NADH can bind. The tetranuclear cluster N3 is coordinated by a four-helix bundle that provides C347$^F$, C350$^F$, and C353$^F$ of the loop connecting helices 1 and 2 of the latter domain, and C393$^F$ of the loop connecting helices 3 and 4 as ligands. N3 is the starting point of an extended electron-transfer chain to ubiquinone, and it is located at an edge-to-edge distance of 7.0 Å from FMN. The model for NuoE comprises 154 amino acids, whereby the first six N-terminal residues were not defined in the electron-density maps. The C-terminal domain of NuoE, comprising residues 80–160 of the model, is highly similar to the thioredoxin-type [2Fe–2S] ferredoxin from *A. aeolicus*[23]. The binuclear cluster N1a located in a hydrophobic environment close to the surface is coordinated by C86$^E$, C91$^E$, C127$^E$, and C131$^E$. The disulfide bond between C144 and C172 found in *T. thermophilus* complex I[24] is not present in the *A. aeolicus* ortholog. Several defined water molecules are found in the NuoE–NuoF interface, possibly supporting electron transfer[25] between N1a and FMN that are located at an edge-to-edge distance of 14.0 Å. In contrast, the longer distance of 19.8 Å between N1a and N3 should disfavor direct electron transfer at a reasonable rate.

NuoEF was reduced by an addition of sodium dithionite solution to a final concentration of 1 mM. The preparation shows the same quaternary structure in both states (Supplementary Table 1). All atoms of the protein in the two states align with a RMSD of 0.41 Å, and only few residues show some conformational flexibility in the different redox states. However, the detailed analysis revealed that a single peptide bond, between E95$^F$ and S96$^F$, undergoes a reversible flip, such that its carbonyl group points toward the flavin in the oxidized state, while it faces the loop coordinating N1a in the reduced state (Fig. 2). This peptide bond is located in the active site of the enzyme between the FMN cofactor and cluster N1a (Fig. 2). In addition, two conserved water molecules are present in the active site in the oxidized state of NuoEF. Upon reduction, one water molecule is replaced by the flipped peptide bond with the carbonyl oxygen as a ligand for the remaining water molecule (Fig. 2). The water oxygen atom now forms a H-bond to the thiol group of C127$^E$, a ligand of N1a, as suggested by its tetrahedral coordination and the S–O distance of 3.1 Å. On the contrary, the position or conformation of the FMN cofactor was not altered by reduction of NuoEF. From this, we tentatively assign the flip of the peptide bond being due to the reduction of N1a.

**Structure of oxidized and reduced NuoEF with nucleotides.** To determine the mode of nucleotide binding, crystals of NuoEF were soaked with NAD$^+$, with NADH and with NADH plus dithionite, leading to refined models at 2.0, 1.8, and 1.9 Å resolution, respectively (Supplementary Table 1). The structures of NADH and NADH plus dithionite-reduced NuoEF were virtually identical. Noteworthy, in the NAD$^+$-bound state, the peptide bond between E95$^F$ and S96$^F$ faces the FMN as in the oxidized protein without nucleotide, while in the NADH-bound state, the peptide bond is pointing toward the loop coordinating N1a as it is in the structure of the reduced protein without nucleotide (Fig. 2). Besides the atoms of ADP-ribose, all atoms of the nicotinamide ring were also clearly resolved in the structures containing NADH, while in most cases only N1 and C4 of the nicotinamide

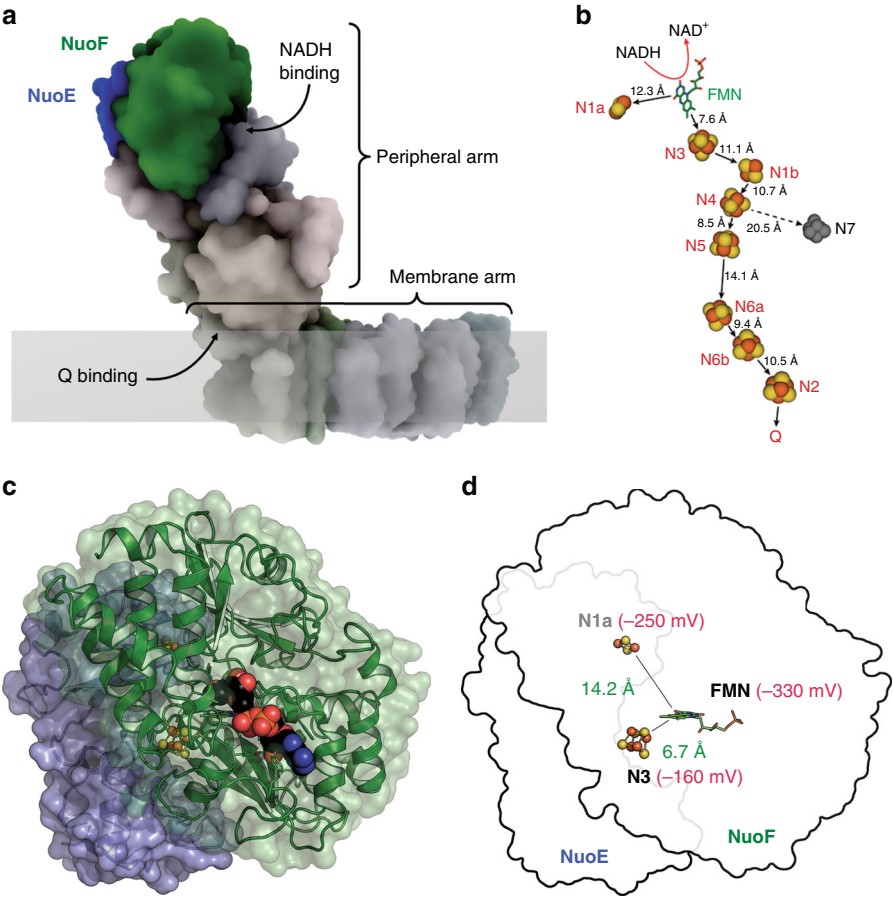

**Fig. 1** The electron input module NuoEF of complex I. **a** Location of subunits NuoE and NuoF in respiratory complex I, at the apical end of the hydrophilic arm. The cartoon is based on the structure of the entire complex I from *T. thermophilus* (PDB code: 4HEA). **b** Schematic representation of the spatial arrangement of the cofactors in the peripheral arm of *T. thermophilus* complex I[24]. The Fe–S clusters involved in the electron transfer reaction are depicted in red/yellow and the edge to edge distance between the cofactors is provided in Å. The distance between N4 and N7 is too long to meet the physiological electron transfer rate. The distance between N2 and the quinone is under discussion. (**c**) Overall structure of the subcomplex NuoEF of *Aquifex aeolicus* complex I. NuoE (blue) is a two-domain protein of 18 kDa with an N-terminal α-helical domain, connected by a short linker sequence to a C-terminal domain with a thioredoxin fold that holds the [2Fe–2S] cluster N1a. The 44 kDa subunit NuoF (green) consists of three domains, of which an antiparallel βαβ domain and a Rossman-fold domain sandwich the cofactor FMN, while a C-terminal α-helical domain holds the [4Fe–4S] cluster N3. **d** Spatial and thermodynamic relationship between the cofactors in NuoEF, with the closest distances between cofactors shown in green, and the midpoint redox potentials in red[22]

moiety are unambiguously detectable in the structures containing $NAD^+$, indicating a dangling of the oxidized nicotinamide moiety along this molecule axis in the binding pocket. Remarkably, the reduced nicotinamide group penetrates significantly deeper into the binding pocket than its oxidized form placing it in an optimal position for hydride transfer (Fig. 3). The amide carbon atom protrudes 1.65 Å deeper into the binding pocket changing its distance to the $C_\alpha$ carbon atom of Glu95$^F$ from 5.37 Å to 4.03 Å. The two positions of the nicotinamide in the binding pocket are here called tight (NADH in reduced NuoEF) and loose ($NAD^+$ in oxidized NuoEF). When bound to reduced NuoEF, the nicotinamide of $NAD^+$ is found very close to the tight position (Fig. 3), suggesting that the orientation of the carbonyl group steers the position of the nicotinamide ring.

The binding of the nucleotides and their interactions with the protein are described in detail in Supplementary Tables 2 and 3. Water molecules that are here resolved for the first time play an important role in substrate binding by forming hydrogen bonds between NADH and NuoF. For example, the nicotinamide N7 is bridged by a water molecule with the side chains of E95$^F$ and D103$^F$. The E97$^F$ side chain binds the NO2' of NADH by a connecting water molecule. One water molecule bridges the

backbone amide of G394$^F$ and the side chain of E184$^F$ with the NO1 of NADH. The new information concerning nucleotide binding as observed in the high-resolution *A. aeolicus* structures is compared with the *T. thermophilus* structure in Supplementary Note.

**Fixing the peptide bond by site-directed mutagenesis**. It cannot be completely ruled out that the redox-dependent structural change in the active site may not be of functional relevance, although this would be quite unusual. We therefore sought to lock the peptide bond in place by exchanging a nearby residue, G129$^E$, with serine and aspartate, respectively. The longer side chains were intended to occupy the position of the bound water molecule that has to make way for the flipping peptide bond according to our structures. The NuoEF variants were produced in *E. coli* (Supplementary Table 4), purified and crystallized. Structures of the oxidized G129S$^E$ and G129D$^E$ variants were refined to 2.4 and 1.9 Å resolution, respectively (Supplementary Table 1). Overall, there were no significant changes from the original NuoEF, with the notable exception of the peptide bond of E95$^F$ that pointed away from FMN in the oxidized state of both variants, as was the case only in the reduced structures of the original protein (Fig. 4).

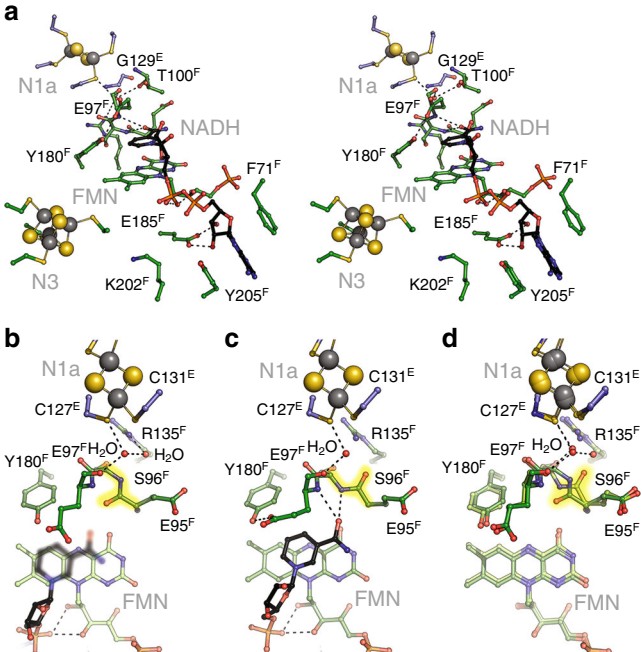

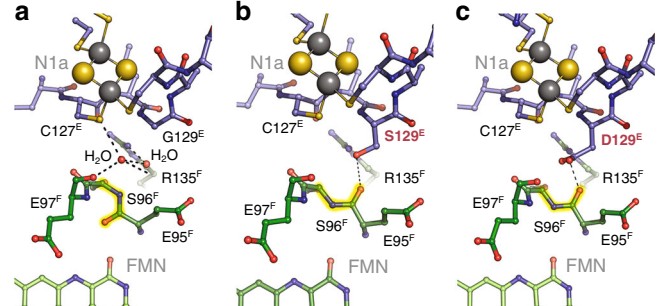

**Fig. 4** Mutagenesis of G129$^E$ influencing the orientation of the peptide bond between E95$^F$ and S96$^F$. **a** In the oxidized form of original NuoEF, the absence of a side chain in amino acid G129$^E$ generates a cavity that harbors two water molecules. The peptide carbonyl of residue E95$^F$ points toward the substrate binding pocket. **b** In the oxidized form of the G129S$^E$ variant, the serine side chain replaces the water molecules, but the peptide bond at E95$^F$ is flipped. The 3.5 Å distance between the hydroxyl group of the serine residue and the carbonyl oxygen indicates a weak hydrogen bond. **c** The even larger side chain in the G129D$^F$ variant has the same effect as in (**b**), but here a stronger H-bond to the flipping peptide carbonyl of 3.2 Å lengths is formed, locking the structure in this configuration. In essence, the oxidized states of the two variants attain a conformation at E95$^F$ that in the original protein is only found in the reduced structure

**Fig. 2** The structure of the NADH-binding site of complex I. **a** Enlarged stereo view into the NADH-binding pocket in the reduced state of the protein and bound NADH (carbon in black). **b** The structure of oxidized NuoEF with bound NAD$^+$, **c** the structure of reduced NuoEF with bound NADH and (**d**) overlay of the structures of oxidized and reduced NuoEF in the absence of pyrimidine nucleotides. The flipping peptide bond is highlighted in yellow and hydrogen-bonding interactions are indicated by dashed lines. The C $=$ O group of the peptide bond between E95$^F$ and S96$^F$ points toward the FMN in the oxidized state of the protein with and without bound NAD$^+$ and flipped away in the reduced state of the protein with and without NADH

Gly/Ser mutation. The same position of the peptide bond was retained when the oxidized G129D$^E$ variant was soaked with NAD$^+$. However, no electron density for the nicotinamide ring and its ribose is detected, while the ADP moiety is well defined. Thus, the nicotinamide might adopt several positions within the binding pocket of the variant. The peptide bond did not change its position when the variant protein was reduced by an addition of dithionite and NADH (Supplementary Fig. 3).

**Fixing the peptide bond disables complex I activity**. In order to assess possible effects on the physiological activity of complex I, the corresponding mutations were introduced in *E. coli* complex I[26]. While E95$^F$ is strictly conserved amongst the different species, S96$^F$ is not (Supplementary Fig. 4). However, S96$^F$ is surrounded by six conserved amino acid residues and the flipping peptide bond is part of a highly conserved region of complex I, in terms of both sequence and structure (Supplementary Fig. 4). To ensure that the methionine residue at the homologous position in the *E. coli* complex does not change the local structure, we generated, purified, and crystallized the *A. aeolicus* S96M$^F$ variant (Supplementary Fig. 4). The mutation did not impose any substantial structural changes, although the hydroxyl group of the original S96 is no longer available for hydrogen bonding to the peptide carbonyl of D94. In addition, the larger side chain of the methionine residue displaces two water molecules, while another water molecule shows up in a different position. Still, the flip of the peptide bond upon reduction of the S96M$^F$ variant is clearly detectable (Fig. 5). Accordingly, statements on the *A. aeolicus* enzyme can be extrapolated to the one from *E. coli*. With the high-sequence identity between *E. coli* and *A. aeolicus* NuoEF, it is unequivocal that G129$^E$ in *A. aeolicus* complex I is homologous to G135$^E$ in *E. coli* and matches its exact function.

The NADH oxidase activity of cytoplasmic membranes of the G135S$^E$ mutant was diminished by approximately 40% in comparison with the original strain, while the G135D$^E$ mutant showed no detectable activity at all (Table 1). Nevertheless, the variants were produced in similar amounts as the original protein as deduced from the NADH/ferricyanide oxidoreductase activity (Table 1). The variants were purified, and the preparations were

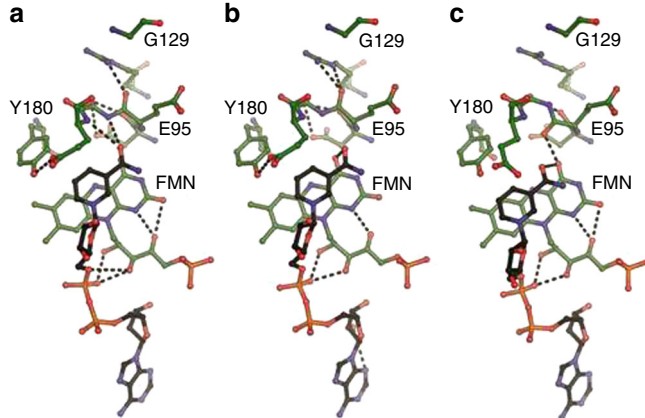

**Fig. 3** Nucleotide binding to the complex I. **a** The structure of the NADH-binding pocket of NuoEF in the reduced state of the protein with bound NADH in the tight position, **b** that of NuoEF in the reduced state of the protein with bound NAD$^+$ close to the tight position and (**c**) that of NuoEF in the oxidized state of the protein with bound NAD$^+$ in the loose position

No electron density for ordered water was observed between position 129$^E$ and the fixed peptide bond in the variant as determined with NuoEF. The 3.5 Å distance between O$_\gamma$ of S129$^E$ and the carbonyl oxygen of E95$^F$ is longer than the 3.2 Å from the carboxylate oxygen of D129$^E$ to this position (Fig. 4). Thus, while both mutations fixed the position of the peptide bond, the Gly/Asp exchange seemed to exert a much stronger effect than the

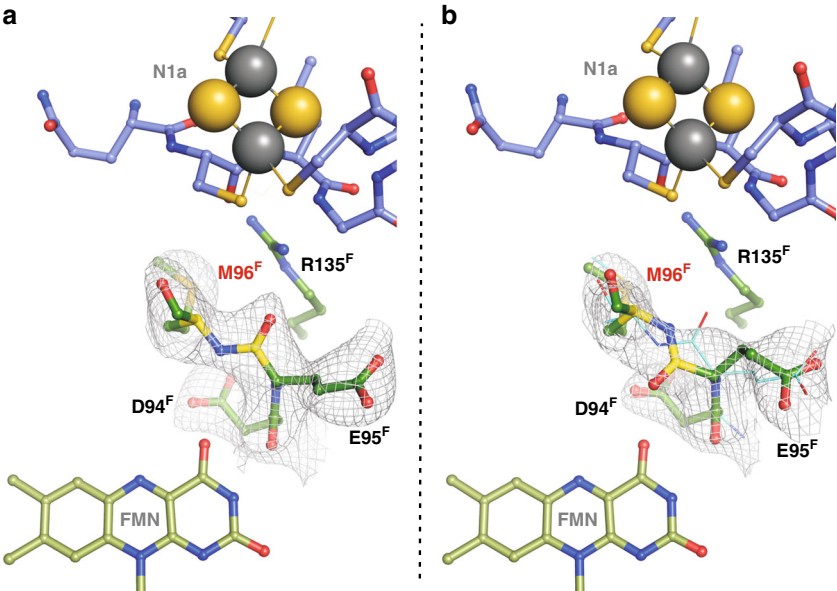

**Fig. 5** The crystal structure of the NADH-binding site of NuoEF S96M$^F$ in the reduced (**a**) and oxidized (**b**) state. Electron densities are given as 2F$_o$–F$_c$ maps contoured at 1.0 σ. **a** One protomer in the asymmetric unit flips the peptide carbonyl of Glu95$^F$ to Arg135$^F$. **b** In the second protomer, the carbonyl predominantly points toward FMN. The according backbone position of the first protomer shown in (**a**) is given as reference (cyan), demonstrating that this conformation does not fit well into the electron density. The crystal structure of the oxidized *A. aeolicus* NuoEF variant S96M that mimics the *E. coli* architecture of the binding pocket shows that here the peptide bond adopts both previously observed positions in each protomer of the asymmetric unit: One corresponds to the position in the oxidized NuoEF, the other to that in the reduced enzyme, although the electron density is not clearly enough defined to assign full occupancies for either conformer due to the relatively low resolution of 3.22 Å (PDB ID 6R7P). Intriguingly, the limited quality of the electron density map unequivocally allows for observing differences between the protomers in preferring a conformer: The first protomer predominantly flips the carbonyl toward Arg135$^F$ (**a**), the other one shows a higher occupancy for the carbonyl flipped toward the FMN (**b**). Thus, the *E. coli* complex I is capable of flipping the peptide bond

**Table 1 Catalytic activity of *E. coli* membranes containing variant forms of complex I**

| Complex I variant | Specific NADH/ ferricyanide oxidoreductase activity$^a$ | | Specific NADH oxidase activity$^b$ | |
|---|---|---|---|---|
| | μmol min$^{-1}$ mg$^{-1}$ | % | μmol min$^{-1}$ mg$^{-1}$ | % |
| Original | 1.6 ± 0.1 | 100 | 0.25 ± 0.02 | 100 |
| G135S$^E$ | 1.3 ± 0.1 | 81 ± 8 | 0.15 ± 0.02 | 60 ± 10 |
| G135D$^E$ | 1.1 ± 0.1 | 69 ± 8 | 0.0 | 0 |

$^a$NADH/ferricyanide oxidoreductase activity is catalyzed solely by the flavin site and was measured in the presence of piericidin A to inhibit quinone reduction. Measurements were conducted at 30 °C in 50 mM MES/OH (pH 6.0), 50 mM NaCl, and 5 mM MgCl$_2$, using 200 μM NADH and 1 mM ferricyanide
$^b$NADH oxidase activity includes the activity of the terminal quinol oxidases. Measurements were conducted at 30 °C in 50 mM MES/OH (pH 6.0), 50 mM NaCl, and 5 mM MgCl$_2$. The reaction was started by an addition of 1 mM NADH
The values reflect complex I activity as the strain lacks the alternative NADH dehydrogenase. The values were derived from three independent measurements

confirmed to contain the subunits of *E. coli* complex I (Supplementary Fig. 5, Supplementary Table 4). Each preparation contained 0.94 ± 0.1 mol FMN/mol protein. The midpoint potential of FMN in the preparations was determined to $E_{m,7} = -318.0 ± 6.0$ mV. The midpoint potential of N1a was not significantly changed from −245 ± 20 mV to −235 ± 20 mV by the mutation. The stability of the flavin site of NuoF was assessed using a "ThermoFAD" protocol[27]. The "melting temperature" of the *E. coli* complex I variants G135S$^E$ and G135D$^E$ were determined to 64 and 63 °C, respectively, and are, thus, very similar to that of the original protein with 69 °C (Supplementary Fig. 6). EPR-spectroscopy confirmed the

presence of all EPR-detectable [Fe–S] clusters (Supplementary Fig. 5). The maximal rate of the NADH/ferricyanide oxidoreductase activity was determined to 540 ± 10 s$^{-1}$ at 30 μM NADH with all preparations. The apparent $K_m^{NADH}$ was determined to 12 μM, respectively, indicating that the mutations did not influence NADH binding. However, the NADH:decyl-ubiquinone oxidoreductase activity of the Ser-variant was about 80% of that of the original complex I, while the Asp-variant showed less than 15% of activity (Table 2). Furthermore, the latter activity faded out within a few minutes during the reaction (Fig. 6). This was not due to a decrease in stability, as the thermal stability of the flavin site was decreased by only 0.8 °C in the preparation of the Asp-variant compared with that of the Ser-variant (Supplementary Fig. 6). The loss of activity by the Asp-variant is even more pronounced in the membrane resulting in a lack of any detectable NADH oxidase activity (Table 1). We attribute the inactivity of the Asp-variant to its incapability to remove the reaction product, NAD$^+$, from the tight binding position (see below).

**The flip of the peptide bond is involved in NAD$^+$ release.** The NADH/ferricyanide oxidoreductase activity of complex I is known to follow a ping–pong–pong mechanism, showing pronounced substrate inhibition[28,29]. The mechanism involves binding of NADH to the complex, electron transfer to FMN, dissociation of NAD$^+$ and sequential binding, and reduction of two molecules of ferricyanide. In standard enzyme assays, the reaction is started by an addition of NADH. However, the ferricyanide-initiated NADH/ferricyanide oxidoreductase activity of *E. coli* complex I is less than 10% of the rate obtained by starting the reaction with NADH[30–32]. This was interpreted as a local structural rearrangement at the NADH-binding site[30,31] or as dissociation of FMN from the reduced enzyme[32]. The latter

**Table 2 Kinetic parameters and H$_2$O$_2$ production by complex I and the variants**

| Complex I variant | NADH:decyl-ubiquinone oxidoreductase activity | $K_{m, app.}$$^{NADH}$ | H$_2$O$_2$ production | |
|---|---|---|---|---|
| | μmol min$^{-1}$ mg$^{-1}$ | μM | nmol min$^{-1}$ mg$^{-1}$ | %$^a$ |
| Original | 3.3 ± 0.1 | 13 ± 1 | 90 ± 8 | 2.7 ± 0.2 |
| G135S$^E$ | 2.6 ± 0.1 | 12 ± 1 | 66 ± 5 | 2.5 ± 0.3 |
| G135D$^E$ | 0.5 ± 0.1 | 13 ± 2 | 11 ± 2 | 2.2 ± 0.4 |
| V96P/N142M$^E$ | 2.7 ± 0.2 | 12 ± 1 | 116 ± 10 | 4.3 ± 0.4 |

$^a$Percentage of the NADH:decyl-ubiquinone oxidoreductase activity
The activities were measured in 50 mM NaCl, 50 mM MES/NaOH, pH 6.0. The initial NADH:decyl-ubiquinone oxidoreductase activity was determined using 5 μg of protein, 60 μM decyl-ubiquinone, and 150 μM NADH. H$_2$O$_2$ production was measured with the Amplex-red assay using 2 U ml$^{-1}$ horseradish peroxidase, 10 μM Amplex-red, and 30 μM NADH. Background rates in the presence of catalase were subtracted. The change in absorbance at 557–620 nm was recorded. Assay conditions: 5 μg of protein, in 1 ml (total volume) 50 mM MES/NaOH, 50 mM NaCl, 0.1% (w/v) dodecyl-maltoside, pH 6.0. The values were derived from three independent measurements

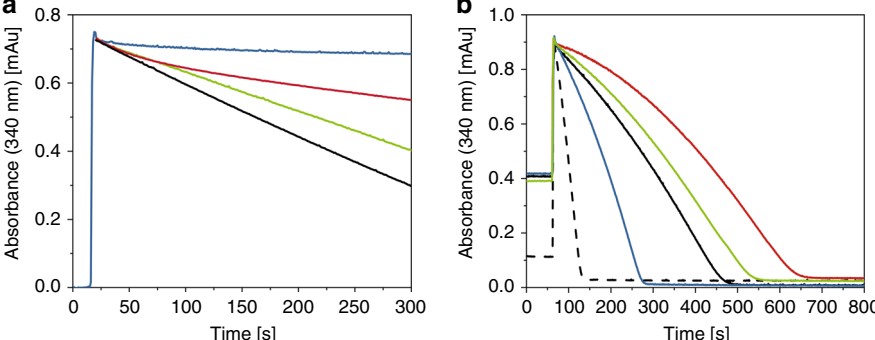

**Fig. 6** NADH-dependent activities of complex I (black) and the G135D$^E$ (red) and the G135S$^E$ (green) variant. **a** NADH:decyl-ubiquinone oxidoreductase activity measured as decrease of the NADH absorbance at 340 nm. The change in absorbance without any protein in the assay is shown in blue. Assay conditions: 5 μg of protein, 60 μM decyl-ubiquinone, 60 μM NADH in 1 ml (total volume) 50 mM MES/NaOH, 50 mM NaCl, 0.1% (w/v) dodecyl-maltoside, pH 6.0. **b** NADH/ferricyanide oxidoreductase activity measured as decrease of the NADH absorbance at 340 nm. The reaction was either started by an addition of 150 μM NADH (dashed line) or 1 mM ferricyanide (solid lines). The rate of the reaction started by NADH addition was 350 ± 10 s$^{-1}$ for all samples. The rates of the reaction started by an addition of ferricyanide were different for all samples and showed a turnover-dependent activation. For comparison, the activity of the V96P/N142M$^E$ variant is shown in blue. Assay conditions: 5 μg of protein in 1 ml (total volume) of the buffer described above. Source data are provided as a Source Data file

proposal is highly unlikely as the FMN cofactor is bound in both, the oxidized and reduced state of NuoEF, with full occupancy, low B-factors and experiences no change in its binding (Figs 2, 3). Furthermore, it was recently shown that the slow initial rate of the ferricyanide-initiated reaction accelerates in a turnover-dependent manner and reaches ~40% of the rate of the NADH-initiated reaction just before the substrate is depleted (Fig. 6). The acceleration of the reaction is interpreted as a slightly faster oxidation of N1a during NADH/ferricyanide oxidoreductase activity than its subsequent reduction by NADH[31]. A detailed kinetic analysis revealed that the slow rate of the ferricyanide-initiated reaction is due to an enhanced binding of the reaction product, NAD$^+$ [31]. Noteworthy, the enhanced binding of NAD$^+$ depends on reduction of N1a, as the ferricyanide-initiated activity was faster and recovered more rapidly with the V96P/N142M$^E$ variant of *E. coli* complex I containing cluster N1a with a 100 mV more negative redox potential of about −390 mV that is hardly reducible by NADH[31,33]. Accordingly, the maximum reaction rate of the V96P/N142M$^E$ variant is twice that of the original protein (Fig. 6).

If now the flip of the peptide bond described above and being determined by the redox state of N1a corresponded to the postulated local structural rearrangement[30,31], the ferricyanide-induced NADH/ferricyanide oxidoreductase activity of the G135D$^E$ variant should be even slower than that of the original protein. Indeed, the variant exhibited a maximum rate of less

than 2% of the reaction started by NADH and showed a longer and pronounced lag in the turnover-dependent activation than the original protein (Fig. 6). A similar kinetic trace was observed with the G135S$^E$ variant. Thus, fixing the flip of the peptide bond of E95$^F$ leads to a slower dissociation of NAD$^+$ from the enzyme further decreasing the rate of the NADH/ferricyanide oxidoreductase activity. It seems therefore highly likely that the flip of the carbonyl group in the active site is a molecular switch to remove the product NAD$^+$ from its tight position in the reduced enzyme triggered by the re-oxidation of N1a. Accordingly, the low and steadily declining NADH:ubiquinone oxidoreductase activity of the G135D$^E$ variant (Fig. 6) is due to product inhibition.

**Binding of NAD$^+$ decreases ROS production.** The production of ROS by complex I depends on the redox state of FMN[8,34]. It is known that the fully reduced flavin is the source of ROS[8]. As the slower dissociation of the reaction product NAD$^+$ prevents binding of the next NADH to further reduce the FMN, we assayed the ROS production by *E. coli* complex I and its G135D$^E$ and G135S$^E$ variants, and compared it with the data published for the V96P/N142M$^E$ variant[31,32] (Table 2). While ROS production by the V96P/N142M$^E$ variant containing the low-potential N1a was found to be 160% increased[31,33], ROS production by the Asp- and Ser-variants was significantly reduced to 12 and 73% of that of the original protein, respectively (Table 2). Thus, bound NAD$^+$

prevents further reduction of the complex and impedes ROS production by hampering further reduction of the complex.

## Discussion

The nicotinamide might occupy a tight or a loose position within the NADH oxidation site of *E. coli* complex I, as identified in the structures of NuoEF with bound NADH and NAD$^+$ (Fig. 3). While all other atoms are well defined, the position of the atoms of the nicotinamide moiety is not easily determined in the electron density maps due to its wobbling in the binding site. However, at 0.5 sigma level, the density of individual atoms is clearly detectable. These atoms were used to model the nicotinamide into the binding site (Fig. 3). While NADH intrudes deeply into the binding pocket of the reduced enzyme (tight), the nicotinamide of NAD$^+$ is bound more peripherally to the oxidized protein (loose, Fig. 3). In NuoEF, the carbonyl group faces cluster N1a in the reduced state enabling binding of the nicotinamide in the tight position. In the oxidized state, the carbonyl group is directed to the nicotinamide pushing it out of the tight position due to electrostatic repulsion with its amide carbonyl (Fig. 3). However, when bound to the reduced enzyme, the NAD$^+$ nicotinamide adopts a position closer to the tight position because the carbonyl group favors the position facing cluster N1a (Fig. 3). This allows the nicotinamide to occupy the tight position although this is a non-productive state of the enzyme/substrate complex (Fig. 7). Thus, the position of the carbonyl group of the flipping E95$^F$–S96$^F$ peptide bond determines the position of the nicotinamide moiety.

The flip of the carbonyl group is regulated by the redox state of N1a because the structural water that is hydrogen bonded to the carbonyl group of the flipping peptide bond in the reduced state is also hydrogen bonded to C127$^E$, a ligand of N1a (Figs 2, 3). In other words, the redox state of N1a regulates NADH oxidation by *E. coli* complex I. We propose that the flipping peptide bond is a molecular switch that supports the release of product NAD$^+$ to prevent this step from becoming rate limiting for the overall electron transfer reaction from NADH to the quinone (Fig. 7). Re-oxidation of reduced N1a leads to a re-orientation of the peptide bond from the cluster toward the nucleotide, thus, pushing NAD$^+$ from the tight to the loose position enabling its dissociation from the active site (Fig. 7).

These findings are in excellent agreement with kinetic data describing the intramolecular electron transfer in *E. coli* complex I[21]. The electron tunneling rates between the clusters of the electron transfer chain from NADH to ubiquinone depend on the redox state of the most distal cluster N2. Reduction of N2 induces conformational changes that decrease the electronic coupling between the clusters and effects the branching between the pathways from reduced FMN to either N2 or N1a (Fig. 1b). When N2 is reduced, the first electron from FMNH$_2$ reduces N1b and N4, while the intermediate FMNH* tends to reduce N1a[21]. Reduced N1a stabilizes binding of NAD$^+$ that in turn prevents NADH binding. This corresponds to an electronic, intramolecular feedback mechanism preventing overreduction of complex I. This is an important feature of the complex because the redox state of FMN determines the rate of ROS production by complex I[8,34]. ROS production by the NuoEF Asp- and Ser-variants was strongly decreased, and the kinetics revealed that the small amount of ROS produced by the Asp-variant originates from its short initial activity phase before the electron transfer reaction slowly faded out. The proposed mechanism enables a simple but fast feedback communication between the reduction state of the quinone pool and the NADH oxidation site that is located in ~100 Å distance from the membrane. In case that the quinone pool is predominantly reduced, it is not capable of oxidizing reduced N2. The reduced state of N2 leads to a reduction of N1a by FMNH*[21]. Thus, a more reduced quinone pool prevents NADH binding. Re-oxidation of the quinone pool leads to a rapid re-oxidation of N1a and subsequent NAD$^+$ release enabling the

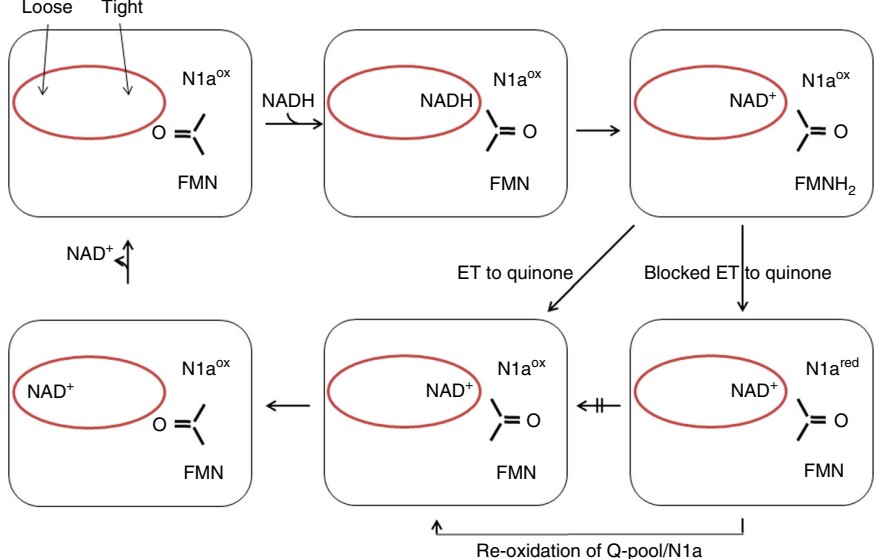

**Fig. 7** NADH oxidation by complex I. The loose- and tight-binding sites for the nicotinamide nucleotide are shown as red sphere, the redox state of the Fe–S cluster N1a and the FMN is indicated, as well as the position of the C = O group of the peptide bond between E95$^F$ and S96$^F$. NADH binds to the tight position of the empty binding site pushing the C = O group towards N1a. Hydride transfer leads to formation of NAD$^+$ in the tight position and of FMNH$_2$. Electron transfer (ET) from FMNH$_2$ to the substrate quinone via the chain of Fe–S clusters leads to the re-oxidation of FMN and the flip of the peptide bond squeezing the NAD$^+$ into the loose position from which it easily dissociates to enable binding of the next NADH. In case that the electrons cannot escape from the complex, e.g., in the presence of a mostly reduced quinone pool, electrons from FMNH$_2$ are kept inside the complex leading to a reduction of N1a[21]. Reduction of N1a prevents the flip of the C = O group resulting in NAD$^+$ bound to the tight position from which it cannot dissociate. This prevents further binding of NADH. Re-oxidation of the quinone pool leads to the oxidation of N1a enabling the release of NAD$^+$

complex to bind the next NADH molecule (Fig. 7). Uncoupling the redox state of N1a from the redox state of N2 as depicted in the V96P/N142M$^E$ variant leads to an enhanced ROS production (Table 2).

The question arises whether the flip of the peptide bond is also a feature of mitochondrial complex I. The flipping carbonyl group of the peptide bond is part of a highly conserved region of the NADH-binding site. The protein backbone from Cys91 to Arg109 (*A. aeolicus*) aligns with an RMSD ranging from 0.42 to 1.36 Å with the homologous region of the bacterial complex from *T. thermophilus* and with that of the complex from ovine, porcine, bovine, mouse, and human mitochondria, respectively, implying a virtually identical structure (Supplementary Fig. 4). Noteworthy, N1a in complex I from *Bos taurus* and *Yarrowia lipolytica* for example is not reducible with NADH[33,35]. This is not simply due to the low redox potential of N1a of –400 mV or less in these organisms[28,36,37] because here, N1a is neither reduced by a strong reductant such as Eu(II)-DTPA[36] nor by NADH when the N1a redox potential is raised to ~ −280 mV by site-directed mutagenesis[33]. However, it may also be speculated that N1a is solely reduced when the electron transfer chain in the mitochondrial complex I adopts a distinct electron distribution when the quinone pool is reduced probably allowing the transient reduction of the cluster. Remarkably, N1a is readily reduced by NADH in a soluble fragment of mitochondrial complex I just containing N1a and N3[33].

## Methods

**Protein purification**. *A. aeolicus* NuoEF and its variants were isolated from an overproducing *E. coli* strain[26] by anion-exchange chromatography on Source 15Q material, affinity chromatography on Ni$^{2+}$-IDA material and size-exclusion chromatography on TSKgel G4000SW in a 50 mM MOPS/NaOH, 50 mM NaCl buffer at pH 7.0[22]. *E. coli* complex I and the corresponding variants were purified from an overproducing strain, as previously described[38]. Briefly, complex I was purified from the detergent extract of membrane proteins by anion-exchange chromatography on Fractogel EMD TMAE Hicap (M) material, affinity chromatography on ProBond Ni$^{2+}$-IDA material, and size-exclusion chromatography on Superose 6 material in 50 mM MES/NaOH, 50 mM NaCl, and 0.1% dodecyl-maltoside, pH 6.0. Typical preparations are listed in Supplementary Table 3.

**Protein crystallization**. The preparation of NuoEF was crystallized at 20 °C by sitting drop vapor diffusion containing 0.1 M Tris/HCl and 1.0 M sodium citrate, pH 6.9–7.1 as precipitant. The crystals were transferred to a 1:1 (v/v) mixture of paratone N/paraffin oil before flash-freezing into liquid nitrogen. For the preparation of the NAD$^+$-complexes, native crystals were soaked in 10 mM NAD$^+$ for 30–60 s before being treated for cryo-protection, as described above. For the preparation of the NADH complexes, native crystals were either soaked with 5 mM NADH or with 5 mM Na-dithionite followed by 5 mM NADH.

**Structure analyses**. Diffraction data of the original NuoEF were collected at beamline X06SA at the Swiss Light Source (Villigen, Switzerland). Crystals were exposed to the X-ray beam at cryogenic temperatures (100 K). The crystal structures in complex with NAD$^+$ and NADH and the structure of the variants were solved by molecular replacement with the native structure as the search model. Supplementary Table 1 summarizes the statistics for crystallographic data collection and structural refinement. Indexing and integration of X-ray diffraction intensities were done using XDS[39] and subsequently scaled and merged using AIMLESS[40] as implemented in autoPROC pipeline[41]. The crystal structure of the NuoEF variants was solved by molecular replacement using MOLREP[42], as implemented within the CCP4 suite. The molecular replacement solution was subjected to a first rigid body refinement using autoBUSTER[43]. Iterative model building was carried out with COOT[44], and all rounds of positional refinement were carried out using autoBUSTER[43] with TLS group definitions and automatically determined NCS symmetry restraints. Ligand dictionaries were generated with the Grade web server (Global Phasing Ltd, Cambridge, England) and their coordinates were placed into 2Fo- Fc electron density maps using AFITT-CL (v.2.1.0; OpenEye Scientific Software, Santa Fe, USA).

**Molecular biology methods**. Plasmids encoding NuoEF and complex I subclones were mutated by using the Quik Change protocol. The subclones containing the site-directed mutations were introduced in the complex I expression vector pBADnuo/nuoF$_{His}$ by λ-red-mediated recombination[26]. The sequences of the mutagenic primers are listed in Supplementary Table 5.

**Enzyme assays**. Enzymatic activities were spectroscopically determined using a TIDAS II diode array spectrometer (J&M, Aalen, Germany)[38]. NADH oxidase activity was determined using a Clark-type electrode (Hansatech, Germany)[38]. EPR spectra were recorded at He-temperature using a Bruker EMX 1/6 spectrometer operating at X-band equipped with an ESR-9 helium-flow cryostat (Bruker, Karlsruhe, Germany)[26]. The redox potential of FMN was determined as described[21]. The stability of the flavin site of NuoF was assessed using a "ThermoFAD" protocol[27]. The fluorescence of free flavin released from heat-denatured complex I and the variants were recorded with an RT-PCR machine (LightCycler 2.0; Roche, Germany) and a home-build machine[45]. The initial temperature of 30 °C was held for 5 min, and then increased by 1 °C every 20 s. To determine the "melting temperature" the first derivative of the original curve was calculated. The midpoint potential of the cofactors was determined by an electrochemical titration using a spectroelectrochemical cell with 0.5-mm pathlength (BASi, Bioanalytical Systems) at 15 °C in an anaerobic tent. Spectra were recorded with a Tidas II diode array UV/vis spectrometer (J&M Analytik, Aalen, Germany). The gold-grid working electrode was modified with 2 mM cysteamine, and mediators were added to accelerate the redox reaction as described[22].

**Reporting summary**. Further information on research design is available in the Nature Research Reporting Summary linked to this article.

## Data availability
The data supporting the findings of this paper are available from the corresponding author upon reasonable request. A reporting summary for this article is available as a Supplementary Information file. Coordinates and structure-factor files have been deposited in the Protein Data Bank, with accession codes 6HL2 (WT, oxidized), 6HL3 (WT, oxidized with NAD$^+$), 6HL4 (WT, reduced with dithionite), 6HLA (WT, reduced with dithionite and NADH), 6HLI (WT, reduced with dithionite plus NAD$^+$), 6Q9C (WT, anaerobically reduced with NADH), 6HLM (variant G129D, oxidized with NAD$^+$), 6Q9G (variant G129D, reduced with NADH), 6HLJ (variant G129S, oxidized with NAD$^+$), 6Q9J (variant G129S, reduced with NADH), 6R7P (variant S96M, oxidized with NAD$^+$), 6Q9K (variant S96M, reduced with NADH). The source data underlying Fig. 6 and Supplementary Figs 5, 6 are provided as a Source Data file.

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

## Acknowledgements

This work was funded by the Deutsche Forschungsgemeinschaft—278002225/RTG 2202 and— 235777276/RTG 1976 and SPP 1927 and the BIOSS Centre for Biological Signalling Studies at Albert-Ludwigs-Universität Freiburg. We thank Lisa Stoss for her excellent help in purifying the S96M$^F$ variant. We thank the beam-line staff at the Swiss Light Source, Villigen, Switzerland, for their excellent assistance with data collection.

## Author contributions

M.S., K.F., S.J., E.G., and R.L. purified the protein, K.F. and S.B. generated the mutants, M.S. and E.G. performed the enzyme kinetic measurements, E.G. performed the electrochemical titration, K.F., S.J., R.L., K.A., and D.F. crystallized the preparations, D.F., S.J., D.W., S.G., and O.E. built and refined the structural models, T.F. performed EPR spectroscopic measurements, analyzed the kinetic data, and designed the project. O.E. and T.F. wrote the paper.

## Additional information

**Competing interests:** The authors declare no competing interests.

