## [Peer Review File · Nature Communications]

Reviewers' Comments:

Reviewer #1:

Remarks to the Author:

The topic of this manuscript is the mechanism of Complex I. This very large membrane-bound protein is part of the respiratory chain where it catalyzes oxidation of NADH and reduction of quinone, linking this reaction to proton pumping across the membrane.

Two closely related topics are discussed in the manuscript: (i) regulation of NADH and NAD⁺ binding and (ii) production of so-called reactive oxygen species (ROS), defined as partly reduced forms of O₂.

Complex I has been shown to be a source of ROS in the respiratory chain, but it is likely that the enzyme has evolved to minimize production of these species. The question is how. The authors suggest here that the mechanism to limit ROS production is intimately linked to regulation of NADH/NAD⁺ binding.

The structural changes observed in this work are relatively minor and it would have been difficult to observe these changes in analyses of the entire complex I. Therefore the authors have worked on an isolated electron input part of complex I and crystallized the protein. This approach enabled the authors to obtain higher resolution than that obtained in structural studies of the entire complex I and to determine structures of several states, i.e. the reduced, oxidized, with bound substrate and product. The data enabled the authors to suggest a mechanism addressing the problems outlined above.

Understanding the functionality of complex I is one of the central problems of Bioenergetics and has implications for medicine because complex I mutations have been linked to specific disease. In addition, understanding mechanisms to minimize ROS production is also of more general medical relevance.

Some specific comments:

1. For the general reader it would be helpful to see a figure in the main part of the manuscript showing the entire peripheral arm where the electron pathway is indicated.
2. p5 l102. How is it known that the flip is reversible?
3. p7, L143. The sentence is unclear. "While unusual..." It is not clear what the word "unusual" refers to. The sentence implies that being unusual it would be of functional relevance. This is difficult to understand. Please explain.
4. The authors discuss flipping of the peptide bond (which determines the binding constant of the substrate/product) in relation to reduction/oxidation of N1a. Consequently, if a mutation is designed to interfere with the flipping, there should be a change in the midpoint potential of N1a by a value that corresponds to the change in the binding constant. Has this been quantified?
5. p11 It is stated that production of ROS is dependent on the redox state of FMN. ROS production by the Asp and Ser variants was reduced. Line 238-240: Why is the possibility of blocked access of dioxygen offered? If NAD⁺ is bound and the complex cannot be reduced, it would be obvious that ROS cannot be formed (?)
6. L249 "deeply"
7. A suggestion is to include a figure showing the mechanism discussed on p. ~12 and to move relevant parts of the text of the Discussion to the figure legend.

Reviewer #2:

Remarks to the Author:

The work entitled "A mechanism to prevent production of reactive oxygen species by *Escherichia coli* respiratory complex I" describes the crystal structures of two subunits (out of 14), NuoE and F from *Aquifex aeolicus* in the oxidized and reduced states and in the presence and absence of NAD⁺. The authors observed a difference in the position of a peptide bond, which seems to correlate with the oxidation state of the close by N1a iron sulfur centre. The position of the referred peptide bond was fixed by site-directed mutagenesis and crystal structures of two mutants in the oxidized state are also presented and discussed. The structures show that in that state the peptide bond in mutants present the orientation of that of the peptide bond in the reduced wild type protein. Possible effects in the physiological activity of complex I were assessed by introducing equivalent mutations that fixed the referred peptide bond in *E. coli* complex I. The authors observed that the mutants presented a decreased rate of reactive oxygen species production and thus suggested the flip of the peptide bond might constitute a mechanism to prevent ROS production.

The suggestion put forward is very interesting, but I have three critical concerns on the correlation between the flip of the peptide bond observed in the structures of *Aquifex aeolicus* NuoEF subunits and the decrease on ROS production observed in *E. coli* complex I.

1- Conservation of the residues establishing the peptide bond. The mentioned flip of the peptide bond is a different orientation of the carbonyl group of the peptide bond between E95 and S96 observed in Nuo F from *Aquifex aeolicus* in the reduced and oxidized states. As shown in Figure S2, S96 is not conserved in *E. coli*, in which equivalent position a methionine is observed. The side chains of a serine and a methionine are very different and one could probably anticipate structural differences in this region.

a) The impact of having a methionine instead of a serine in the structure has to be addressed. Can the authors be certain that the carbonyl of the peptide bond in a protein with a methionine instead of a serine would behave in the same way?

b) Possibly the crystal structure of S96M NuoF mutant would help to clarify this point.

With the available data it is hard to understand that the observed structural features in NuoEF subunits from *A. aeolicus* can be extrapolated to *E. coli* complex I.

2- Fixing the flexible peptide bond. The authors obtained structures of NuoEF mutants from *A. aeolicus* fixing the flexible peptide bond. These were obtained in the oxidized state and were similar to the structure of the wild-type NuoEF in the reduced state and thus the authors concluded that the bond was "fixed".

a) What would happen to the bond in the mutants in the reduced state? One could hypothesize that reduction of N1a might alter the hydrogen network and the peptide bond might adopt a different orientation. Evidences that the peptide was really fixed are not present, since structures in only the oxidation state are shown.

b) Again, even if the bond is fixed for the *A. aeolicus* proteins it does not necessarily mean that it would also be fixed in the case of *E. coli* enzyme, because the proton network in the region is certainly different due to the differences between a serine and a methionine side chain.

The data presented cannot exclude that conformational changes of the "fixed" bond may occur upon reduction.

3- Even considering that the referred above extrapolation is possible to make and that the bond could be really "fixed", there is another major concern. The authors mentioned that the ROS

production by the mutants in *E. coli* complex I is reduced in relation to the wild type. However, the activity of the complex, NADH:decylubiquinone oxidoreductase is also decreased in the mutants. Remarkably, as reported in table 2, the relation between the NADH:decylubiquinone oxidoreductase activity and ROS production is similar for the wild type and the mutants, which indicates that the decrease in ROS production correlates with the decrease of the activity. If fixing the peptide bond would interfere with the mechanism to prevent ROS production then one would expect that ROS production would be more affected than the activity, which is not the case. The results show that the mutations affect the NADH:decylubiquinone oxidoreductase and thus also ROS production. There is no evidence that the mutations indicate the existence of a mechanism to prevent ROS production.

Additional points

4- Line 113 – Structure of the oxidized and reduced NuoEF with bound nucleotides. The structures of the reduced NuoEF with NAD⁺ and NADH were obtained by soaking the dithionite-reduced NuoEF with each nucleotide. Was soaking of the oxidized protein with NADH tried? The authors should discuss the possible effects of reducing the protein only with NADH.

5- Lines 194 - The flip of the peptide bond is involved in NAD⁺ release. Figure 5. Please include the data for G121S mutant in the two panels. Please use arrows to indicate the different additions in both panels. The results obtained for G121S mutant should be discussed in the text.

Reviewer #3:

Remarks to the Author:

Nature Communications: Schulte, M.,.....Friedrich, T.

In this interesting manuscript the authors have provided three excellent resolution x-ray crystal structures of the catalytically active soluble NuoE-NuoF fragment of complex I from *Escherichia coli*. Interestingly a single peptide bond between Glu95 and Ser96 seems to flip when the enzyme is in its oxidized versus reduced state, i.e., the peptide bond points towards the FMN molecule when oxidized, whereas, it flips and points towards Fe-S cluster N1a when the enzyme is reduced. The authors conclude that this flip of the peptide bond along with displacement of a water molecule acts as a molecular switch. The study provides interesting insight into the mechanism of *E. coli* complex I in how cluster N1a participates in reducing the formation of reactive oxygen species (ROS) by the enzyme. The paper is likely to be of interest to those working in bioenergetics and in particular of complex I which is certainly an active area of investigation.

Specific comments:

1. On page 6, line 124-126 the authors state that "Remarkably, the reduced nicotinamide group penetrates significantly deeper into the binding pocket than in its oxidized form..." From a static two dimensional figure (Figure 3) this is hard to see. How much deeper is the nicotinamide? What distance are we talking about? It would help if the authors could be more specific.
2. Some distances are shown in Supplementary Table S2, however, pages 11 and 12 of the supplementary material are both labeled as interactions of NADH with NuoEF in the reduced state. However, page 12, seems to indicate that NAD⁺ is present rather than NADH. Is this dithionite reduced enzyme with NAD⁺ present, or what? There is no information in the Table S2 to clarify this.
3. In Figure 4 showing the oxidized wild type and variant proteins it is stated in the Figure 4 legend that in 4c there is a "direct H-bond" to the flipping peptide carbonyl. The way the legend is written it implies that in G129S that there is not a direct H-bond between Ser129 and the S96 peptide carbonyl, however, in Fig. 4b the dotted line suggests a H-bond is present? Is the peptide carbonyl "locked" in both the Ser129 and Asp129 variants? This is not clear as shown or written. The section on page 7, lines 153-158 explain this more clearly so it seems that the figure legend

should be modified to better reflect the distance variations 3.2 to 3.5 angstroms in the two variants.

4. The last paragraph of the Discussion (page 13, lines 299-308) asks whether the ‘flip of the peptide bond’ is a relevant feature of mitochondrial complex I? It is not stated, however, what sort of structural conservation there is in this region of the flavin and N1a between the *E. coli* and bovine or *Yarrowia lipolytica* enzyme. Is it even possible that this is the case, i.e., is a Glu-Ser conserved in the equivalent region in the mitochondrial proteins? It would help the reader if this was mentioned.

5. Although the x-ray structures seem of high quality do the authors have any evidence that Fe-S center N1a does not become reduced by the x-ray beam in what they are calling the oxidized structures? If so this might be briefly mentioned.

Reviewer #1 (Remarks to the Author):

The topic of this manuscript is the mechanism of Complex I. This very large membrane-bound protein is part of the respiratory chain where it catalyzes oxidation of NADH and reduction of quinone, linking this reaction to proton pumping across the membrane.

Two closely related topics are discussed in the manuscript: (i) regulation of NADH and NAD⁺ binding and (ii) production of so-called reactive oxygen species (ROS), defined as partly reduced forms of O₂.

Complex I has been shown to be a source of ROS in the respiratory chain, but it is likely that the enzyme has evolved to minimize production of these species. The question is how. The authors suggest here that the mechanism to limit ROS production is intimately linked to regulation of NADH/NAD⁺ binding.

The structural changes observed in this work are relatively minor and it would have been difficult to observe these changes in analyses of the entire complex I. Therefore the authors have worked on an isolated electron input part of complex I and crystallized the protein. This approach enabled the authors to obtain higher resolution than that obtained in structural studies of the entire complex I and to determine structures of several states, i.e. the reduced, oxidized, with bound substrate and product. The data enabled the authors to suggest a mechanism addressing the problems outlined above.

Understanding the functionality of complex I is one of the central problems of Bioenergetics and has implications for medicine because complex I mutations have been linked to specific disease. In addition, understanding mechanisms to minimize ROS production is also of more general medical relevance.

A: We thank the reviewer for these very supportive comments.

Some specific comments:

1. For the general reader it would be helpful to see a figure in the main part of the manuscript showing the entire peripheral arm where the electron pathway is indicated.

A: We have moved the scheme from Figure S1 showing the entire electron pathway in the peripheral arm of the complex to Figure 1 of the main text (as Fig. 1b).

2. p5 l102. How is it known that the flip is reversible?

A: This is concluded from the enzyme kinetics of the variant proteins. Fixing the flip by the G129S^E and G129D^E mutation results either in a strongly decreased activity or in an inactive enzyme (Tables 1 and 2; Figure 5). As we see the different positions of the peptide bond in the structures of the reduced and oxidized NuoEF, the flip has to be reversible in the original protein. If it would be irreversible, the mutations would have no effect.

3. p7, L143. The sentence is unclear. "While unusual..." It is not clear what the word "unusual" refers to. The sentence implies that being unusual it would be of functional relevance. This is difficult to understand. Please explain.

A: The sentence has been rephrased to: 'It cannot be completely ruled out that the redox-dependent structural change in the active site may not be of functional relevance, although this would be quite unusual.' (l. 144/145)

4. The authors discuss flipping of the peptide bond (which determines the binding constant of the substrate/product) in relation to reduction/oxidation of N1a. Consequently, if a mutation is designed to interfere with the flipping, there should be a change in the midpoint potential of N1a by a value that corresponds to the change in the binding constant. Has this been quantified?

A: This is an excellent proposal that we tested by electrochemical titration. It turned out that the mutation did not alter the midpoint potential of N1a (WT: -254 ± 20 mV; G129D^E: -235 ± 20 mV). This is now mentioned in the text (lines 179 - 180). The electrochemical titration is now described in the 'Online Methods' section (lines 480 - 485).

5. p11 It is stated that production of ROS is dependent on the redox state of FMN. ROS production by the Asp and Ser variants was reduced. Line 238-240: Why is the possibility of blocked access of dioxygen offered? If NAD⁺ is bound and the complex cannot be reduced, it would be obvious that ROS cannot be formed (?)

A: We fully agree with this statement and removed the alternative proposal that bound NAD⁺ blocks the access of dioxygen to the enzyme. (now lines 245 - 246)

6. L249 "deeply"

A: Has been corrected (now line 255).

7. A suggestion is to include a figure showing the mechanism discussed on p. ~12 and to move relevant parts of the text of the Discussion to the figure legend.

A: A scheme describing the mechanism is now included as new Figure 6. A major part of the text of the discussion was moved to the figure legend.

Reviewer #2 (Remarks to the Author):

The work entitled "A mechanism to prevent production of reactive oxygen species by Escherichia coli respiratory complex I" describes the crystal structures of two subunits (out of 14), NuoE and F from Aquifex aeolicus in the oxidized and reduced state and in the presence and absence of NAD⁺. The authors observed a difference in the position of a peptide bond, which seems to correlate with the oxidation state of the close by N1a iron sulfur centre. The position of the referred peptide bond was fixed by site-directed mutagenesis and crystal structures of two mutants in the oxidized state are also presented and discussed. The structures show that in that state the peptide bond in mutants present the orientation of that of the peptide bond in the reduced wild type protein. Possible effects in the physiological activity of complex I were assessed by introducing equivalent mutations that

fixed the referred peptide bond in *E. coli* complex I. The authors observed that the mutants presented a decreased rate of reactive oxygen species production and thus suggested the flip of the peptide bond might constitute a mechanism to prevent ROS production.

The suggestion put forward is very interesting, but I have three critical concerns on the correlation between the flip of the peptide bond observed in the structures of *Aquifex aeolicus* NuoEF subunits and the decrease on ROS production observed in *E. coli* complex I.

1- Conservation of the residues establishing the peptide bond. The mentioned flip of the peptide bond is a different orientation of the carbonyl group of the peptide bond between E95 and S96 observed in Nuo F from *Aquifex aeolicus* in the reduced and oxidized states. As shown in Figure S2, S96 is not conserved in *E. coli*, in which equivalent position a methionine is observed. The side chains of a serine and a methionine are very different and one could probably anticipate structural differences in this region.

a) The impact of having a methionine instead of a serine in the structure has to be addressed. Can the authors be certain that the carbonyl of the peptide bond in a protein with a methionine instead of a serine would behave in the same way?

*A: The 'flipping' carbonyl group of the peptide bond is part of a highly conserved region of the protein, namely the NADH binding site. We used the published structures of the complex from *Thermus thermophilus* and from the complex of ovine, porcine, bovine, mouse and human mitochondria and overlaid the region containing the peptide bond (see below). The structure of the complex from *Yarrowia lipolytica* could not be included into this comparison, because this region is structurally not defined (pdb-code: 4WZ7).*

*Superposition of the protein backbone from the complex from various organisms of the regions containing the loop. The position of the FMN is shown and the structures were aligned relative to the isoalloxazine ring of the FMN. The region from Cys91^F to Arg109^F (*A. aeolicus* numbering) is shown.*

The rmsd between the individual structures ranges from 0.42 to 1.36 Å, the mean value is 0.7 Å. Thus, the structures are considered to be identical. From this we expect that introducing a methionine (E. coli) instead of a serine residue (A. aeolicus) to will not have any effect. Furthermore, this region is highly conserved by judging from the amino acid sequence. Position 96 (marked in yellow) is not conserved but it is surrounded by six conserved amino acids:

Bovine	KYLVVNADE G	EPGTCKDREI	IRHDPH
Mouse	KYLVVNADE G	EPGTCKDREI	MRHDPH
Rat	KYLVVNADE G	EPGTCKDREI	MRHDPH
Human	KYLVVNADE G	EPGTCKDREI	IRHDPH
Y. lipolytica	RYLVVNADE G	EPGTCKDREI	MRKDPH
Paracoccus	SYLVINADES	EPATCKDREI	MRHDPH
E. coli	RYLLCNADE M	EPGTYKDRLI	MEQLPH
T. thermophilus	HYLICNADES	EPGSFKDRII	LEDVPH
A. aeolicus	RYFICNADES	EPGTFKDRII	IERDPH
Consensus	* ****	** ***	**

Accordingly, it is most likely that the change from a serine residue to a methionine residue will not change the structure and the data obtained with A. aeolicus can be extrapolated without any problems to the E. coli complex.

b) Possibly the crystal structure of S96M NuoF mutant would help to clarify this point.

With the available data it is hard to understand that the observed structural features in NuoEF subunits from A. aeolicus can be extrapolated to E. coli complex I.

A: To check the above statement that the data obtained with A. aeolicus can be extrapolated to the E. coli complex, we generated the S96M^F variant as proposed. From the structure it is evident that in contrast to what has been speculated by the reviewer the mutation did not impose any structural difference in the protein. An H-bond from Ser96 to the peptide carbonyl from Asp94 is lacking due to the lack of the hydroxyl group. Furthermore, due to its larger size, the methionine residue displaces two water molecules while another one shows up in a different position:

Structure of NuoEF original protein (blue) and the S96M^F variant (orange). The water molecules displaced by the methionine residue are circled in blue, the water molecule not present in the original protein is circled in orange. The position of the other amino acid side chain did not change.

Because the mutation did not change the position of any other amino acid side chain it is clear that the data obtained with A. aeolicus can be extrapolated to those obtained with E. coli and vice versa. We do not think that the structure of the S96M^F variant contributes to the understanding of the mechanism to prevent ROS production proposed in the manuscript. Therefore, we would like to not include the structure into the manuscript. However, the structure is mentioned in Supplementary Table S1 and will be deposited in the pdb.

2- Fixing the flexible peptide bond. The authors obtained structures of NuoEF mutants from A. aeolicus fixing the flexible peptide bond. These were obtained in the oxidized state and were similar to the structure of the wild-type NuoEF in the reduced state and thus the authors concluded that the bond was “fixed”.

a) What would happen to the bond in the mutants in the reduced state? One could hypothesize that reduction of N1a might alter the hydrogen network and the peptide bond might adopt a different orientation. Evidences that the peptide was really fixed are not present, since structures in only the oxidation state are shown.

A: We added a new figure to the Supplementary Material to answer this question (Supplementary Fig. S3). We have determined the structures of both, the G129D^E and the G129S^E variants in the reduced and in the oxidized state. The position of the peptide bond is in exactly the same position in the reduced and in the oxidized state of both variants as expected. This now stated in lines 163 - 165.

b) Again, even if the bond is fixed for the A. aeolicus proteins it does not necessary mean that it would also be fixed in the case of E. coli enzyme, because the proton network in the region is certainly different due to the differences between a serine and a methionine side chain.

A : As stated above, we have experimentally and unequivocally shown that there are no differences in the proton network due to the mutation from a serine to a methionine residue. This proves that the flipping peptide bond experiences the same interactions in A. aeolicus and E. coli NuoEF.

The data presented cannot exclude that conformational changes of the “fixed” bond may occur upon reduction.

A: The flipping peptide bond has a high conformational flexibility. Notably, the G129D^E mutation leads to a dead enzyme (Tab. 1) providing very strong evidence that the conformational flexibility of this bond is strongly, if not fully restricted in the variant. The G129S^E mutation leads to a decrease of 40% of the activity (Tab. 1). Here, the mutation strongly enhances the probability to locate the peptide bond in the position facing the Fe/S cluster N1a and not the FMN.

3- Even considering that the referred above extrapolation is possible to make and that the bond could be really “fixed”, there is another major concern. The authors mentioned that the ROS production by the mutants in E. coli complex I is reduced in relation to the wild type. However, the activity of the complex, NADH:decylubiquinone oxidoreductase is also decreased in the mutants.

Remarkably, as reported in table 2, the relation between the NADH:decylubiquinone oxidoreductase activity and ROS production is similar for the wild type and the mutants, which indicates that the decrease in ROS production correlates with the decrease of the activity. If fixing the peptide bond would interfere with the mechanism to prevent ROS production then one would expect that ROS production would be more affected than the activity, which is not the case.

The results show that the mutations affect the NADH:decylubiquinone oxidoreductase and thus also ROS production. There is no evidence that the mutations indicate the existence of a mechanism to prevent ROS production.

A: This description is partly correct (first paragraph) but incomplete. The major finding is that there are two different binding sites for the nicotinamide nucleotide, the tight and the loose site. Externally added NAD⁺ binds to the loose site from which it can easily dissociate. NAD⁺ derived from the oxidation of NADH is bound to the tight site from which it can hardly dissociate. To translocate NAD⁺ from the tight to the loose site, the flip of the peptide bond is needed. However, the flip of the peptide bond is regulated by the redox state of Fe-S cluster N1a that is reduced only when the most distal cluster of the chain, N2, is reduced (ref. 21). In turn, N2 is reduced only when the quinone pool is mainly in the reduced state as well. Thus, the flip of the peptide bonds reports the redox state of the quinone pool to the NADH binding site that is in about 100 Å distance. In fact, by blocking the NADH binding site, not only ROS production is inhibited but also the NADH:decyl-ubiquinone oxidoreductase activity is equally throttled. Inevitably, the inhibition arises from a malfunctional ejection mechanism of NAD⁺ from the tight site that is otherwise regulated by redox chemistry. Externally added NAD⁺ has no effect on the NADH:decyl-ubiquinone activity and on ROS production. The mutations result in variants lacking the regulation mechanism. In these variants, ROS production is blocked because NADH is unable to feed electrons into the enzyme due to the presence of NAD⁺ in the tight site. Blocking the NADH binding site by NAD⁺ also blocks NADH:decyl-ubiquinone oxidoreductase activity. The inhibition is reversible because a re-oxidation of the quinone-pool leads to a re-oxidation of N2 and subsequently of N1a so that NAD⁺ can be released from the tight site. This regulation is lacking in the variants. The newly described mechanism is the reversible and regulated block of the NADH binding site by NAD⁺ being bound to the tight site.

Additional points

4- Line 113 – Structure of the oxidized and reduced NuoEF with bound nucleotides. The structures of the reduced NuoEF with NAD⁺ and NADH were obtained by soaking the dithionite-reduced NuoEF with each nucleotide. Was soaking of the oxidized protein with NADH tried? The authors should discuss the possible effects of reducing the protein only with NADH.

A: When we submitted the manuscript we had solved both the structure of NuoEF reduced by dithionite and NADH and the one reduced reduced by NADH alone. As the resolution of the structure obtained by reduction with dithionite and NADH was initially much better than the one obtained with NADH alone, we used the first one to describe the structure and the changes due to the redox state of the protein. In the meantime we obtained a structure of NuoEF reduced by NADH at 1.8 Å which is better than the resolution obtained by an addition of dithionite and NADH (1.9 Å). The structure of NuoEF reduced by NADH is now shown in Supplementary Table S1 and is now deposited in the pdb. The structure of NuoEF reduced by NADH and the one reduced by dithionite and NADH are virtually identical.

5- Lines 194 - The flip of the peptide bond is involved in NAD⁺ release. Figure 5. Please include the

data for G121S mutant in the two panels. Please use arrows to indicate the different additions in both panels. The results obtained for G121S mutant should be discussed in the text.

A: The effect of the G121S^E mutation on the release of NAD⁺ is similar to that of the G129D^E mutation but not as pronounced. Thus, the kinetics obtained with the G121S^E provide no new information. In order to not confuse the reader with an overburdened figure, we would like to keep Fig. 5 as it is. The effect of this mutation on the activity is now stated in the text (lines 227 - 228).

Reviewer #3 (Remarks to the Author):

Nature Communications: Schulte, M.,.....Friedrich, T.

In this interesting manuscript the authors have provided three excellent resolution x-ray crystal structures of the catalytically active soluble NuoE-NuoF fragment of complex I from Escherichia coli. Interestingly a single peptide bond between Glu95 and Ser96 seems to flip when the enzyme is in its oxidized versus reduced state, i.e., the peptide bond points towards the FMN molecule when oxidized, whereas, it flips and points towards Fe-S cluster N1a when the enzyme is reduced. The authors conclude that this flip of the peptide bond along with displacement of a water molecule acts as a molecular switch. The study provides interesting insight into the mechanism of E. coli complex I in how cluster N1a participates in reducing the formation of reactive oxygen species (ROS) by the enzyme. The paper is likely to be of interest to those working in bioenergetics and in particular of complex I which is certainly an active area of investigation.

A: We thank the reviewer for these very supportive comments.

Specific comments:

1. On page 6, line 124-126 the authors state that "Remarkably, the reduced nicotinamide group penetrates significantly deeper into the binding pocket than in its oxidized form..." From a static two dimensional figure (Figure 3) this is hard to see. How much deeper is the nicotinamide? What distance are we talking about? It would help if the authors could be more specific.

A: We thank the reviewer for this most helpful comment. The movement of the nicotinamide is now described by numbers in lines 126/127: 'The amide carbon atom protrudes 1.65 Å deeper into the binding pocket changing its distance to the C_α carbon atom of Glu95^F from 5.37 Å to 4.03 Å.'

2. Some distances are shown in Supplementary Table S2, however, pages 11 and 12 of the supplementary material are both labeled as interactions of NADH with NuoEF in the reduced state. However, page 12, seems to indicate that NAD⁺ is present rather than NADH. Is this dithionite reduced enzyme with NAD⁺ present, or what? There is no information in the Table S2 to clarify this.

A: We are very sorry for causing any confusion. The tables have been corrected. Table S2 on p. 11 of the Supplementary Material shows distances within the preparation of NuoEF reduced with dithionite and NADH. Table S2 on p. 12 shows interactions of NAD⁺ with the oxidized NuoEF. To clarify this point, we split the tables into Tab. S2: Interactions of NADH with NuoEF reduced by dithionite and NADH and Tab. S3: Interactions of NAD⁺ with NuoEF in the oxidized state.

3. In Figure 4 showing the oxidized wild type and variant proteins it is stated in the Figure 4 legend that in 4c there is a “direct H-bond” to the flipping peptide carbonyl. The way the legend is written it implies that in G129S that there is not a direct H-bond between Ser129 and the S96 peptide carbonyl, however, in Fig. 4b the dotted line suggests a H-bond is present? Is the peptide carbonyl “locked” in both the Ser129 and Asp129 variants? This is not clear as shown or written. The section on page 7, lines 153-158 explain this more clearly so it seems that the figure legend should be modified to better reflect the distance variations 3.2 to 3.5 angstroms in the two variants.

A: We fully agree with this statement. The legend to Figure 4 has been modified accordingly to ‘In the oxidized form of the G129S^E variant, the serine side chain replaces the water molecules, but the peptide bond at E95^F is flipped. The 3.5 Å distance between the hydroxyl group of the serine residue and the carbonyl oxygen indicates a weak hydrogen bond. (c) The even larger side chain in the G129D^F variant has the same effect as in (b), but here a stronger H-bond to the flipping peptide carbonyl of 3.2 Å lengths is formed, locking the structure in this configuration. (lines 373 - 377)

4. The last paragraph of the Discussion (page 13, lines 299-308) asks whether the ‘flip of the peptide bond’ is a relevant feature of mitochondrial complex I? It is not stated, however, what sort of structural conservation there is in this region of the flavin and N1a between the E. coli and bovine or Yarrowia lipolytica enzyme. Is it even possible that this is the case, i.e., is a Glu-Ser conserved in the equivalent region in the mitochondrial proteins? It would help the reader if this was mentioned.

A: The relevant region of the protein, namely the specific part of the NADH binding site is highly conserved as judged by its primary and tertiary structure (see answer to the comments of reviewer 2, point 1). This is now included in the ‘Discussion’ as ‘The flipping carbonyl group of the peptide bond is part of a highly conserved region of the NADH binding site. The protein backbone from Cys91 to Arg109 (A. aeolicus) aligns with an RMSD ranging from 0.42 to 1.36 Å with the homologous region of the bacterial complex from T. thermophilus and with that of the mitochondrial complex from ovine, porcine, bovine, mouse and human, respectively, implying a virtually identical structure. (lines 299 - 303).

5. Although the x-ray structures seem of high quality do the authors have any evidence that Fe-S center N1a does not become reduced by the x-ray beam in what they are calling the oxidized structures? If so this might be briefly mentioned.

A: Crystals were exposed to the x-ray beam at cryogenic temperatures (100 K). Under this condition the Fe-S clusters in particular those with a negative redox potential are not reduced as has been described (e.g. Einsle et al. JACS 2007, 129, 2210-2211; Spatzal et al. Nat. Commun. 2016, 7:10902). This is now mentioned in ‘Online Methods’. (lines 450 - 451).

Reviewers' Comments:

Reviewer #1:

Remarks to the Author:

The authors have addressed my comments.

Reviewer #2:

Remarks to the Author:

The authors have addressed the points raised, namely they produced the crucial mutant S96M NuoF and obtained its structure in the dith + NADH reduced form. Nevertheless, several points still bring concern.

1- Conservation of the residues establishing the peptide bond.

a) The presence of a Met in the E. coli complex in the equivalent position of that of S96 NuoF of A. aeolicus has to be mentioned in the main text.

b) The structure of the NuoF S96M has to be described. This is crucial to all the discussion because it validates the extrapolation that is done from the observations in the structures of NuoEF from A. aeolicus to the functional results of complex I from E. coli. Please present figures with the same perspective as figures 1 and 4.

2- Flexibility of the peptide bond in the NuoF S96M mutant. The structure of the NuoF S96M mutant was only obtained in its dith + NADH reduced form. The peptide bond is in the same position as those of the dith + NADH reduced wt. But it is also in the same position of that of the mutants both in the oxidized and reduced forms, coincident with the position in which the peptide bond is "locked". Is there evidence that in the NuoF S96M mutant the E95-M96 peptide bond is flexible? This issue has to be discussed in the main text.

3- Mechanism to prevent production of ROS. Again, the authors mentioned that the ROS production by the mutants in E. coli complex I is reduced in relation to the wild type. However, the activity of the complex, NADH:decylubiquinone oxidoreductase is also decreased in the mutants. In fact the ration (%) of the NADH:decylubiquinone oxidoreductase and ROS production is very similar between the WT and the mutants (table 2, last column). This means that these two activities could not be dissociated. The presented V96P/N142ME mutant is very illustrative. It shows that ROS production increased while NADH:decylubiquinone oxidoreductase decreased. In this case it can be concluded that the mutation affected differently/independently the two reactions. In the case of the mutants G135S and G135D it can be concluded that these impact on the NADH:decylubiquinone oxidoreductase, but there is no direct evidence that the mutations impact directly/independently on ROS production. This has to be addressed in the main text. And consequently the title should be rephrased because it is misleading.

Additional points

4- Line 113 – Structure of the oxidized and reduced NuoEF with bound nucleotides. It is very nice the authors have obtained the structure of the reduced form of NuoEF by NADH alone. This is most informative and should be the one discussed here instead of the structure of the reduced form of NuoEF by dith + NADH. The structure of the reduced form of NuoEF by NADH alone is more informative because i) it is the physiological relevant one and ii) has a higher resolution.

5- Lines 194 - The flip of the peptide bond is involved in NAD⁺ release. Figure 5. Please include the data for G121S mutant in the two panels. The fig can be drawn in order not to be overburdened to the reader and it will allow the reader to interpret the data by himself.

Reviewer #3:

Remarks to the Author:

I feel the authors have responded appropriately to the reviews.

Reviewer #1 (Remarks to the Author):

The authors have addressed my comments.

A: We thank the reviewer for this positive statement.

Reviewer #2 (Remarks to the Author):

The authors have addressed the points raised, namely they produced the crucial mutant S96M NuoF and obtained its structure in the dith + NADH reduced form. Nevertheless, several points still bring concern.

1- Conservation of the residues establishing the peptide bond.

a) The presence of a Met in the E. coli complex in the equivalent position of that of S96 NuoF of A. aeolicus has to be mentioned in the main text.

b) The structure of the NuoF S96M has to be described. This is crucial to all the discussion because it validates the extrapolation that is done from the observations in the structures of NuoEF from A. aeolicus to the functional results of complex I from E. coli. Please present figures with the same perspective as figures 1 and 4.

A: We fully agree with the reviewer. Accordingly, a completely new paragraph has been introduced on p. 8, lines 170 - 182. Here, the position 96 of NuoF, the structure of the S96M^F variant and the flip of the peptide bond in this variant are described in detail. Within this paragraph a reference is made to the newly introduced Supplementary Fig. 4 that shows the sequence comparison in the region containing the flipping peptide bond, the structural conservation of the entire fold and the structure of the S96M^F variant in comparison to the original protein as well as the two positions of the peptide bond in the reduced and oxidized state of the S96M^F variant from the same perspective as provided in Figs. 2 and 3 that show the flip of the peptide bond.

2- Flexibility of the peptide bond in the NuoF S96M mutant. The structure of the NuoF S96M mutant was only obtained in its dith + NADH reduced form. The peptide bond is in the same position as those of the dith + NADH reduced wt. But it is also in the same position of that of the mutants both in the oxidized and reduced forms, coincident with the position in which the peptide bond is “locked”. Is there evidence that in the NuoF S96M mutant the E95-M96 peptide bond is flexible? This issue has to be discussed in the main text.

A: We thank the reviewer for this comment. Despite some effort to obtain a high resolution structure of the oxidized S96M^F variant by checking more than 50 crystals from different preparations and different crystallization batches, we did not obtain a resolution better than 3.22 Å. Noteworthy, the obtained electron density was sufficiently detailed for a robust analysis. The new structure is now included in Supplementary Figure 4 and described and compared to the structure of the reduced protein in the new paragraph on p. 8, lines 170 - 182. The according data collection and refinement statistics are included in Supplementary Tab. 1. The structure of the oxidized variant disclosed that the peptide bond adopts both previously observed positions in each protomer of the asymmetric unit: One corresponds to the position in the oxidized NuoEF, the other to that in the reduced NuoEF. When modeling the electron density with only one of the two positions, a residual electron density is clearly

visible demonstrating the presence of different states. Although this new structure has been obtained at considerably lower resolution as compared to the other structures presented in our manuscript, the electron density map unequivocally allows for observing differences between the protomers in preferring a conformer: The first protomer predominantly flips the carbonyl toward the FMN, the other one shows a higher occupancy for the carbonyl flipped toward Arg135^F. Thus, the E. coli homologue is capable of flipping the peptide bond. We fully agree with the reviewer that these data are important in order to justify the extrapolation from structural to functional data. However, we disagree that this should belong to the main text. In our opinion, the main text should contain the central statement of the research paper i.e. the detection of the flipping peptide bond and its role for the mechanism of respiratory complex I. The structural validation that both systems (A. aeolicus and E. coli) are equivalent is important and has been included now in the main text in on p. 8, lines 170 - 182 but is in its details a more technical issue that would dilute the information presented in the main text. In order to not distract the reader from the main statement we would like to keep the figures obtained from the S96M^F variant in 'Supplementary Material'.

3- Mechanism to prevent production of ROS. Again, the authors mentioned that the ROS production by the mutants in E. coli complex I is reduced in relation to the wild type. However, the activity of the complex, NADH:decylubiquinone oxidoreductase is also decreased in the mutants. In fact the ration (%) of the NADH:decylubiquinone oxidoreductase and ROS production is very similar between the WT and the mutants (table 2, last column). This means that these two activities could not be dissociated. The presented V96P/N142ME mutant is very illustrative. It shows that ROS production increased while NADH:decylubiquinone oxidoreductase decreased. In this case it can be concluded that the mutation affected differently/independently the two reactions. In the case of the mutants G135S and G135D it can be concluded that these impact on the NADH:decylubiquinone oxidoreductase, but there is no direct evidence that the mutations impact directly/independently on ROS production. This has to be addressed in the main text. And consequently the title should be rephrased because it is misleading.

A: The clear statement of the manuscript is that NAD⁺ bound in the tight position blocks the electron input site of complex I, thus preventing a further oxidation of NADH. By doing so, the NADH:decylubiquinone oxidoreductase activity is inevitably blocked as is the production of reactive oxygen species (ROS). In line with this argument, it is evident that NADH binding is needed for both activities. Hence, these processes cannot be dissected, as we have clearly stated in our manuscript (and cited by the reviewer). Accordingly, we have not proposed that the two activities can be 'dissociated'. Rather, the important finding is that this physical block is implemented by the redox state of a Fe/S cluster changing the conformation of a peptide bond. Hence, the block of the NADH binding site is a) reversible and b) regulated by the redox state of the enzyme (and finally by the redox state of the quinone pool). When comparing the effects of G135D^E and the V96P/N142M^E mutants, the reviewer may have missed the fact that the two mutations affect different properties of the enzyme: With the G135D^E mutation we fixed the peptide bond. Thus, the enzyme has lost its ability to push NAD⁺ out of the tight binding site. This is a structural change interfering with substrate binding. In the V96P/N142M^E variant, by contrast, the architecture of the substrate binding site is untouched, while here, the midpoint potential of the Fe/S cluster is changed. Here, it can no longer be reduced by NADH. This is a change of the thermodynamic property of a cofactor that ultimately interferes with electron tunneling. Accordingly, the reduced NADH:decylubiquinone oxidoreductase activity of the

V96P/N142M^F variant is due to a slightly disturbed (20%) internal electron transfer process as described in our original publication (Birrell et al. (2013) Biochem. J., our reference 33). Furthermore, this data fully supports our proposal that in the original protein, where the Fe/S cluster can be reduced by NADH and where the peptide bond in question is capable of flipping, this flip helps to diminish ROS production. In the V96P/N142M^F variant, the flip is still possible, however, the ejection of NAD⁺ out of the tight binding site is no longer functional, because now uncoupled from the redox state of the complex. Thus, electrons may now enter the reduced complex I and the surplus electrons residing on FMN may produce ROS (our references 8 and 34). Taken together, the reduced oxidoreductase activity of the V96P/N142M^F variant is due to a diminished intramolecular electron tunneling in the first place, while the enhanced ROS production is a side effect due to an 'over-reduction' of the complex. This is in full agreement with our hypothesis. Consequently, we are not willing to rephrase the title of the manuscript.

Additional points

4- Line 113 – Structure of the oxidized and reduced NuoEF with bound nucleotides. It is very nice the authors have obtained the structure of the reduced form of NuoEF by NADH alone. This is most informative and should be the one discussed here instead of the structure of the reduced form of NuoEF by dith + NADH. The structure of the reduced form of NuoEF by NADH alone is more informative because i) it is the physiological relevant one and ii) has a higher resolution.

A: The reviewer is right here and we took this into account. However, because the structure of NuoEF reduced with NADH is identical to the one of NuoEF reduced by NADH plus dithionite, we did not have to change the corresponding figures. Instead, we quote in the text that the corresponding structures were obtained from NuoEF reduced by NADH. In addition, we remark that an identical structure was obtained by first reducing the crystals with dithionite and sub-sequential soaking with NADH (p. 6, lines 114 - 117).

5- Lines 194 - The flip of the peptide bond is involved in NAD⁺ release. Figure 5. Please include the data for G121S mutant in the two panels. The fig can be drawn in order not to be overburdened to the reader and it will allow the reader to interpret the data by himself.

A: We apologize that we do not understand this statement. First, we do not describe a G121S mutant in our manuscript. Second, Fig. 5 shows the activities of the original NuoEF and the G135D^E variant. Hence, we conclude that the reviewer refers to the G135S^E variant. Indeed, data on this variant are not shown in this figure. Instead, melting curves of the G135D^E and G135S^E variants, provided in Supplementary Fig. S5 (now S6), are referred to in line 194. The phrase 'The flip of the peptide bond is involved in NAD⁺ release' the reviewer quotes in his revision as being found in l. 194, however, is found in line 194 only in the unrevised manuscript but not in the revised version. Thus, we are wondering whether the reviewer is indeed referring to the revised version of the manuscript. It might be possible, the reviewer refers to Fig. 4 of the original manuscript rather than Fig. 5 of the revised version. But here, the data for the two G135D^E and G135S^E variants are already included. Furthermore, in the revised version, we added a new Supplementary Fig. 3 showing the structures of these two variants in the oxidized and in the reduced state, all from the same perspective. Still, we

are not aware of a figure showing data obtained from the variants that has two panels. Accordingly, we are afraid, we may not be able to satisfyingly answer the request of the reviewer.

Reviewer #3 (Remarks to the Author):

I feel the authors have responded appropriately to the reviews.

A: We thank the reviewer for this positive statement.

Reviewers' Comments:

Reviewer #2:

Remarks to the Author:

The authors have addressed the points raised. However, I still have two minor concerns;

1- Flexibility of the peptide bond in the NuoF S96M mutant. Please include the structural data, both the reduced and oxidized form of the mutant in the main text, because it validates the extrapolation that is done from the observations in the structures of NuoEF from *A. aeolicus* to the functional results of complex I from *E. coli*.

2- Fig 5. Please include the data concerning G135S variant.

Reviewer #2 (Remarks to the Author):

The authors have addressed the points raised. However, I still have two minor concerns;

*1- Flexibility of the peptide bond in the NuoF S96M mutant. Please include the structural data, both the reduced and oxidized form of the mutant in the main text, because it validates the extrapolation that is done from the observations in the structures of NuoEF from *A. aeolicus* to the functional results of complex I from *E. coli*.*

The reduced and oxidized forms of the S96M^E variant are now discussed in the main manuscript and the structural data are shown as new Figure 5. The corresponding data have been removed from the Supplementary Information.

2- Fig 5. Please include the data concerning G135S variant.

The kinetic data obtained with G135S^E variant are now included in Figure 6 (formerly Figure 5).